# Ambient Air Pollution and Risk of Admission Due to Asthma in the Three Largest Urban Agglomerations in Poland: A Time-Stratified, Case-Crossover Study

**DOI:** 10.3390/ijerph19105988

**Published:** 2022-05-14

**Authors:** Piotr Dąbrowiecki, Andrzej Chciałowski, Agata Dąbrowiecka, Artur Badyda

**Affiliations:** 1Department of Allergology and Infectious Diseases, Military Institute of Medicine, 04-141 Warsaw, Poland; pdabrowiecki@wim.mil.pl (P.D.); achcialowski@wim.mil.pl (A.C.); 2Polish Federation of Asthma Allergy and COPD Patients Associations, 01-604 Warsaw, Poland; 3Maria Skłodowska-Curie Medical School, 00-136 Warsaw, Poland; agata.dabrowiecka123@gmail.com; 4Faculty of Building Services, Hydro and Environmental Engineering, Warsaw University of Technology, 00-653 Warsaw, Poland

**Keywords:** asthma, air pollution, PM_10_, PM_2.5_, nitrogen dioxide, sulfur dioxide, distributed lag nonlinear models

## Abstract

Ambient air pollution in urban areas may trigger asthma exacerbations. We carried out a time-series analysis of the association between the concentrations of various air pollutants and the risk of hospital admission due to asthma over 7 days from exposure. We used distributed lag nonlinear models to analyze data gathered between 2010 and 2018 in the three largest urban agglomerations in Poland. Overall, there were 31,919 asthma hospitalizations. Over 7 days since exposure, the rate ratio (95%CI) for admission per 10 µg/m^3^ was 1.013 (1.002–1.024) for PM_10_; 1.014 (1.000–1.028) for PM_2.5_; 1.054 (1.031–1.078) for NO_2_; and 1.044 for SO_2_ (95%CI: 0.986–1.104). For all pollutants, the risk of admission was the greatest on the day of exposure (day 0), decreased below baseline on days 1 and 2, and then increased gradually up to day 6. The proportions (95%CI) of hospitalizations attributable to air pollution were 4.52% (0.80%–8.14%) for PM_10_; 3.74% (0.29%–7.11%) for PM_2.5_; 16.4% (10.0%–21.8%) for NO_2_; and 2.50% (−0.75%–5.36%) for SO_2_. In conclusion, PM_2.5_, PM_10_, NO_2_, and SO_2_ pollution was associated with an increased risk of hospital admission due to asthma in the three largest urban agglomerations in Poland over nine years.

## 1. Introduction

Asthma is one of the most common chronic diseases among children and adults worldwide, with an increasing prevalence. In 2016, around 340 million people worldwide suffered from asthma, with asthma being a leading cause of disease burden (23.7 million disability-adjusted life years globally) [1,2]. Asthma is a respiratory disease caused by genetic and environmental factors that leads to airway constriction and inflammation [3]. The course of asthma is characterized by intermittent exacerbations that range from mild, self-limiting symptoms to severe dyspnoea requiring urgent medical care in a hospital [4,5]. Hospital admissions may thus be viewed as a proxy for both severe asthma exacerbations and the costs related to healthcare system utilization [6].

Ambient air pollution in urban areas is an established trigger for asthma exacerbations. Several air pollutants are found in increased concentrations in urban areas due to emissions from traffic, municipal, and household sources, and also from power generation and industry. These include the particulate matter of an aerodynamic diameter no greater than 2.5 μm or 10 μm (PM_2.5_, PM_10_), nitrogen dioxide (NO_2_), or sulfur dioxide (SO_2_), all of which can induce airway constriction and airway inflammation, the two key aspects of asthma pathogenesis [7]. A recent meta-analysis of 84 studies showed that the risk of asthma exacerbations at lags 0 or 1 day increased with increasing ambient air concentrations of PM_2.5_, PM_10_, NO_2_, and SO_2_ [8].

Air quality in Polish cities is among the worst in Europe, particularly regarding PM and polycyclic aromatic hydrocarbons; however, pollution with NO_2_ and ozone is increasing as well. Twenty-three of the fifty European cities with the highest ambient levels of PM_2.5_ are in Poland [9]. The sources of pollution include heating-related emissions from individual households during the winter months and the increasing burden of road traffic. In some cities, emissions from the industrial sector are substantial [10]. However, no study to date has investigated the relationship between urban air pollution and the risk of hospital admission due to asthma in a sample representative for the whole of Poland [11,12]. Moreover, data on the lagged effects of air pollutants on the risk of hospital admission due to asthma are lacking. In this study, we gathered data from the three largest urban areas in Poland and used a time-stratified case-crossover design to analyze how ambient air pollution was related to the risk of hospital admission due to asthma over several days since exposure.

## 2. Materials and Methods

### 2.1. Setting

This study was a time-series analysis of the association between ambient air pollution and the risk of hospital admission due to asthma during the period between 1 January 2010 and 31 December 2018 in the three largest urban agglomerations in Poland: Tricity, Warsaw, and Cracow, with about 8.7% of the total population in Poland residing in these agglomerations. Tricity (population 0.75 million) is a metropolitan area consisting of Gdansk, Sopot, and Gdynia, and is located in northern Poland, on the Baltic coast. Warsaw (1.8 million) lies in east-central Poland, in the Masovian Lowland, about 260 km from the Baltic Sea. Cracow (0.77 million) lies in southern Poland, about 300 km south of Warsaw, at the foot of the Carpathian Mountains, 219 m above sea level. Appendix A shows the geographic location of the three agglomerations. The study was a retrospective analysis of publicly available data and therefore ethical approval was not needed.

### 2.2. Definition and Measurements

Admission due to asthma was defined as a hospitalization billed with the J45 or J46 codes from the International Classification of Disease 10th Edition as the main reason for admission. The counts of daily admissions were gathered from the National Health Fund, i.e., the only public insurer financing the universal healthcare system in Poland and encompassed the admissions to all publicly financed hospitals in the three cities during the study period. In Poland, because of the universal healthcare system, publicly financed hospitalizations represent nearly all hospitalizations as out-of-pocket financing of inpatient care is negligible. The daily counts of admissions did not encompass emergency room visits.

The mean daily concentrations of PM_10_, PM_2.5_, NO_2_, and SO_2_ and the mean daily values of relative humidity, wind velocity, precipitation, and ambient air pressure were taken from the measuring stations of the Chief Inspectorate of Environmental Protection (https://powietrze.gios.gov.pl/pjp/maps/measuringstation accessed on 27 October 2021). For each city, we averaged the values from all available stations.

### 2.3. Statistical Analysis

We used a time-stratified case-crossover design that compared exposures on case days with exposures on control days in the same calendar month of the same year. This design allowed the examination of how differences in exposure contributed to differences in counts of admissions. Moreover, the design allowed the pooling of data from the three cities because control periods were specific for the cities. The counts were modeled with conditional quasi-Poisson regression, which enables an analysis of aggregated exposure data and the control of overdispersion [13]. We used a stratum variable of “city-year-month” to control for a long-term trend and seasonality and to pool data from the three cities [14,15]. In addition, we adjusted the models for the effect of the day of the week. The logarithm of the total population of each city was used as the offset variable. We used distributed lag nonlinear models (DLNM) to investigate the delayed associations of PM_10_, PM_2.5_, NO_2_, and SO_2_ with admission risk. DLNMs are based on the function of cross-basis, which enables simultaneous analysis of the risk of an event along the dimensions of exposure and lag [16]. The daily concentrations of air pollutants were analyzed in separate models over 7 days from exposure, with a linear relationship between the concentration of a pollutant and the risk of admission, whereas the lag was modeled with a polynomial function and 4 degrees of freedom (df). We analyzed the risk of admission over 7 days since exposure because the effects of air pollution on asthma exacerbations seem unlikely beyond this period. The models were adjusted for the effect of meteorological variables, i.e., temperature, relative humidity, and atmospheric pressure, with natural cubic splines (df = 3) and a lag of up to one day. We used the “group” option in the “crossbasis” function in the “dlnm” package to select individual cities [17]. The results were presented as rate ratios (RR) per 10 μg/m^3^ for each day after exposure, including cumulative effects. Moreover, we calculated the RRs for the overall effects within the 6-day period and the percentages of hospitalizations attributable to the individual air pollutants over this period [18]. The “gnm” package was used to fit the conditional quasi-Poisson models. The “FluMoDL” package was used to calculate the percentages of hospitalizations attributable to air pollution. Results were considered statistically significant when 95% confidence intervals (CI) did not span the value of no effect (1.0 for RR; 0 for percentage of attributable admissions). All calculations were completed in R (version 3.6.2).

## 3. Results

### 3.1. Air Pollution and Asthma Hospitalizations

Figure 1 shows the daily concentrations of PM_10_, PM_2.5_ NO_2_, and SO_2_ for the entire study period, with increased concentrations during the winter months. The median concentrations of PM_10_, PM_2.5_, NO_2,_ and SO_2_ were the greatest in Cracow and were the lowest in Tricity (Table 1). The values of all meteorological variables were similar across cities (Table 1).

In total, there were 31,919 hospitalizations due to the J45 or J46 codes in the three cities (details in Table 1). Figure 1E shows the monthly counts of admissions due to the J45 or J46 codes, with increased counts during the winter months, mirroring the temporal pattern for air pollutants.

### 3.2. Lagged Associations between PM_10_, PM_2_._5,_ NO_2_, and SO_2_ and the Risk of Admission

PM_10_ concentration was associated with the greatest risk of admission on the day of exposure; the risk decreased below baseline on days 1 and 2 but then increased up to day 6 (Figure 2A). The cumulative risk of admission over 7 days since exposure was U-shaped (Figure 2E). Overall, during 7 days since exposure, an increase of 10 μg/m^3^ in PM_10_ concentration was associated with an increased admission risk (RR per 10 μg/m^3^ = 1.013; 95%CI: 1.002–1.024), whereas 4.52% (95%CI: 0.80%, 8.14%) of all asthma admissions were attributable to PM_10_. Figure 2I and Figure 3A show the association between PM_10_ and the risk of admission for the range of exposures and lags. The results were similar for PM_2.5_: RR was 1.014 (1.000–1.028) and the percentage of attributable admission was 3.74% (0.29–7.11%; see Figure 2B,F,J and Figure 3B).

The temporal pattern between exposure and the risk of admission was similar for NO_2_, with the greatest risk of admission on the day of exposure, a reduced risk on days 2–3, and a subsequent increase up to day 6 (Figure 2C). The cumulative risk of admission over 7 days since exposure for NO_2_ was U-shaped and significantly increased on each day since exposure (Figure 2G). Overall, during 7 days since exposure, an increase in NO_2_ was associated with an increased admission risk (RR per 10 μg/m^3^ = 1.054; 95%CI: 1.031–1.078), whereas 16.4% (95%CI: 10.0%–21.8%) of all asthma admissions were attributable to NO_2_. Figure 2K and Figure 3C show the association between NO_2_ and the risk of admission for the range of exposures and lags.

The temporal pattern between SO_2_ exposure and the risk of admission was similar to that for PM and NO_2_, but the changes in risk were non-significant (Figure 2D,H,L). Overall, during 7 days since exposure, an increase in SO_2_ was associated with an increased admission risk (RR per 10 μg/m^3^ = 1.044; 95%CI: 0.986–1.104), whereas 2.50% (95%CI: −0.75–5.36%) of all asthma admissions were attributable to SO_2_. Figure 2I and Figure 3D show the effect of SO_2_ on the risk of admission for the range of exposures and lags.

## 4. Discussion

This study showed that increased concentrations of urban air pollutants were associated with an increased short-term risk of hospital admission due to asthma in the three largest agglomerations in Poland. Descriptive data already suggested that the changes in the concentrations of air pollutants were paralleled by asthma admission counts (Figure 1). Further analyses confirmed that exposure to air pollutants was associated with an increased risk of hospital admission due to asthma. The risk was the greatest for NO_2_, lower for PM_10_ and PM_2.5_, and non-significant for SO_2_. For all pollutants, we observed “harvesting”, with the greatest risk on the day of exposure, followed by a decrease in risk below baseline on days 2 and 3, and a further increase up to day 6. Considerable proportions of asthma hospitalizations were attributable to air pollution. The lack of a significant association between the concentration of SO_2_ and asthma admission could be due to low overall concentrations of this pollutant (below harmful levels).

Our findings are in line with previous reports. In a study that used a case-crossover design to analyze the risk of outpatient hospital visits for asthma in 17 Chinese cities, Lu et al. found that the risk was significantly increased by exposure to PM_10_, PM_2.5_, NO_2_, and SO_2_ in ambient air [19]. In that study, the lagged effects for PM_10_, PM_2.5_, and NO_2_ were similar to those in our study, with the greatest risk on the day of exposure, followed by a second peak on day 3, whereas for SO_2_ the risk was the greatest one day after exposure and then it returned to baseline [19]. Moreover, the overall effects of air pollution on the risk of an outpatient hospital visit due to an asthma exacerbation over 5 days since exposure in the study by Lu et al. were similar to our current report (RR = 1.004 for PM_2.5_, 1.005 for PM_10_, 1.030 for NO_2_, and 1.015 for SO_2_) [19], although we analyzed the frequency of inpatient hospitalizations. Likewise, the overall RR per 10 μg/m^3^ for PM_10_, PM_2.5_, NO_2_, and, SO_2_ in our analyses were consistent with those reported previously in studies from Hongkong (1.034, 1.028, 1.019, 1.021) [20] and Shijiazhuang (1.005, 1.007, 1.016, 1.020) [21]. In four European cities (APHEA project), an increase in NO_2_ concentration was significantly associated with the risk of emergency asthma admissions (RR, 1.029 per 50 μg/m^3^; PM_10_ was not assessed; the overall effect of SO_2_ was non-significant) [22]. In general, our results are in line with a meta-analysis of 87 studies, in which, at different lags, the risk of emergency room visits or hospitalizations due to asthma was significantly increased by exposure to PM_10_ (RR. 1.010 per 10 μg/m^3^) and NO_2_ (1.018 per 10 μg/m^3^) [23].

In our study, larger effects were observed for NO_2_ than for PM_2.5_, PM_10_, and SO_2_, which is in line with some previous studies. In the study by Lu et al. and our study, the greatest percentage of asthma admissions was attributable to NO_2_ (10.9%, ~16% in our study), lower for PM_10_ and PM_2.5_ (4.2%, ~4% in our study), and the lowest for SO_2_ (3.7%, ~2% in our study) [19]. Similarly, in the study by Ca et al., an interquartile range increase in the concentrations of PM_10_, SO_2_, and NO_2_ was associated with an increase in asthma hospitalizations by 1.82%, 6.41%, and 8.26%, respectively [24]. In contrast, in Shiraz, Iran, an interquartile range increase in PM_10_ was associated with a greater increase in the number of asthma hospitalizations (31%) compared to NO_2_ (11%) and SO_2_ (22%) [25]. This “reversed” effect in Shiraz could be due to occasional dust storms increasing PM_10_ concentrations in addition to urban emissions [26]. The link between SO_2_ and asthma exacerbations may be less evident than for NO_2_ or PM [27]. For example, in the previously mentioned APHEA study, SO_2_ exposure significantly increased the risk of an emergency admission due to asthma in children only [22]. In our study, SO_2_ pollution was associated with an increased risk of asthma admission, although this effect was non-significant. In contrast, in a study carried out in Alberta, Canada, SO_2_ exposure was associated with a decreased risk of an emergency room visit due to asthma, whereas NO_2_ and PM_10_ exposure increased the risk [28].

The greatest risk of admission due to asthma for all pollutants in our study was found on the day of exposure, with a subsequent risk reduction below baseline on days 2 and 3, followed by an increase up to day 6. This pattern is consistent with the phenomenon of “harvesting”, which has been observed for the effect of air pollution on mortality [29,30]. Harvesting could reflect a pattern in which air pollution affects vulnerable people immediately, who are then admitted to hospital on the day of exposure. These people are likely to stay in hospital or die shortly after admission and thus the risk remains decreased on subsequent days because the most susceptible population is smaller. The effect of air pollution on less susceptible people might be seen later, with the risk increasing gradually. In addition to an epidemiological explanation of harvesting, the two-phase effect of air pollutants on asthma admissions could be because of direct irritation of the airways on the day of exposure, causing immediate bronchoconstriction; then, a gradual pro-inflammatory effect in the airways could further worsen asthma symptoms [7]. Some previous studies analyzing the effect of air pollutants on the risk of asthma exacerbations found harvesting [31,32,33,34], whereas others did not find [19,35]. Lu et al. observed that the risk of asthma admission was the greatest on the day of exposure for PM_2.5_, PM_10_, and NO_2_, but it did not decrease below baseline on subsequent days. Similarly, Lee et al. found that the risk of emergency room visits due to asthma increased over 3 days since exposure to PM_10_, NO_2_, and SO_2_ [35]. However, Lu et al. analyzed outpatient hospital admissions, Lee et al. analyzed emergency room visits, whereas we analyzed inpatient admissions, which most likely represent more severe asthma exacerbations. In Poland, patients with symptoms that do not require immediate admission to hospital may be treated in emergency departments for up to 3 days; if they improve, patients are discharged home and the emergency room visit does not count as a hospitalization.

The limitations of our study need to be acknowledged. We did not analyze how demographic data could modify the effects of air pollutants on the risk of asthma admission. This was because we could obtain only aggregated data on the counts of admissions, whereas demographic data were unavailable. Therefore, we analyzed pooled data from children and adults and the effects of air pollution on asthma exacerbations could differ by age group. Moreover, we could not differentiate between repeated visits by the same patient. The data on air pollutants were taken from a few measuring stations; however, individualized data would be more accurate. Finally, the data on asthma admissions did not include mild and moderate exacerbations, which account for a substantial proportion of asthma exacerbations. The strengths of our study include an analysis of an urban population representative for Poland (nearly 10% of the country’s population) and the use of a time-stratified case-crossover design that reduces confounding by conditioning data specific to location and a short time window [13,14]. The pooling of data from the three cities was justified because of the similarities in climates and ethnicity of populations across the cities. Moreover, the data on the number of admissions and pollutant concentrations were taken from the same institutions for all cities.

## 5. Conclusions

In conclusion, PM_2.5_, PM_10_, NO_2_, and SO_2_ pollution was associated with an increased risk of hospital admission due to asthma in the three largest urban agglomerations in Poland over nine years. Our results add further evidence on the association between air pollution and the burden of respiratory diseases in Poland [36,37]. We hope this accumulating evidence will help push forward initiatives to reduce emissions from individual households in Poland, the main source of PM, and to reduce NO_2_ emissions, which are mainly attributable to increasing traffic [38].

## Figures and Tables

**Figure 1 ijerph-19-05988-f001:**
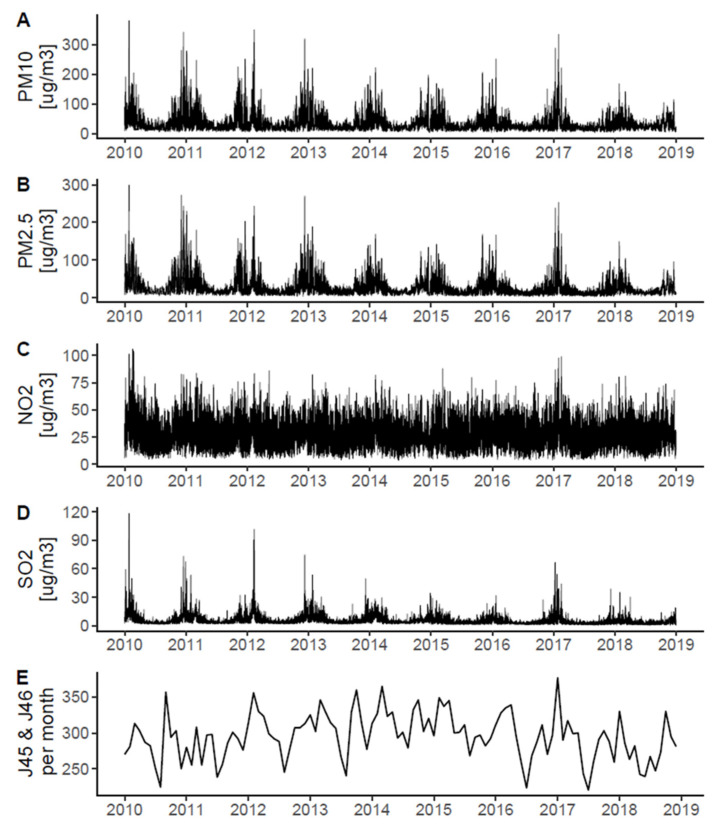
Daily concentrations of PM_10_ (**A**), PM_2.5_ (**B**), NO_2_ (**C**), and SO_2_ (**D**) in the study period. Monthly counts of hospitalizations due to the J45 or J46 ICD-10 codes (**E**) (“asthma”).

**Figure 2 ijerph-19-05988-f002:**
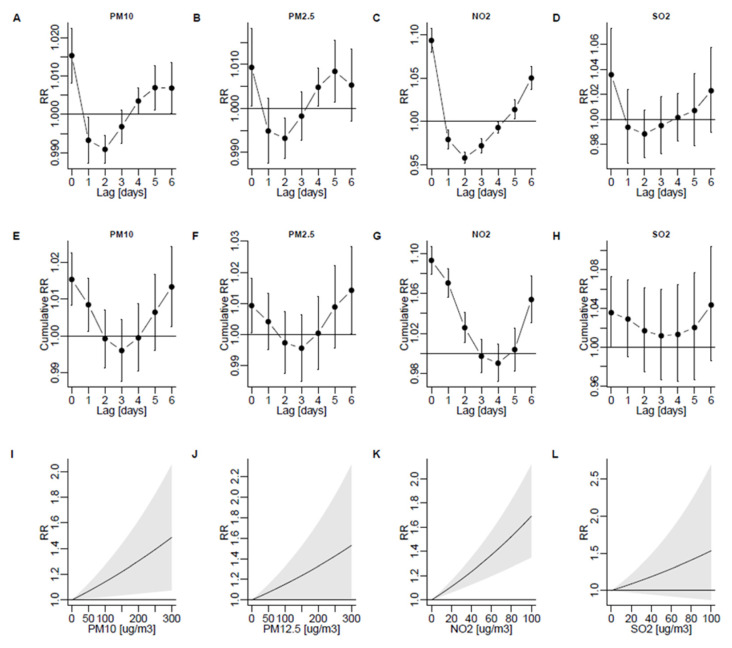
Risk of admission due to J45 or J46 (“asthma”) per 10 μg/m^3^ for PM_10_ (**A**), PM_2.5_ (**B**) NO_2_ (**C**), and SO_2_ (**D**) depending on lag. Cumulative effects are shown in (**E**–**H**). Predicted admission risk within a range of exposure is shown in (**I**–**L**). 95% confidence intervals are shown as bars (**A**–**F**) or shaded areas (**G**–**I**). RR, rate ratio.

**Figure 3 ijerph-19-05988-f003:**
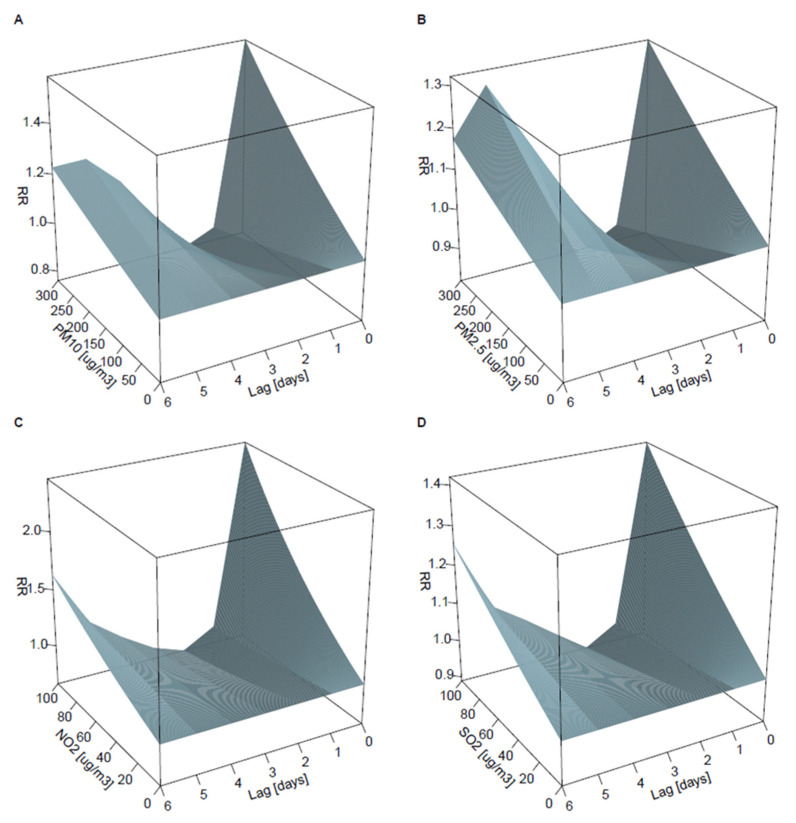
Risk for admission due to J45 or J46 (“asthma”) depending on time since exposure (lag) and concentrations of PM_10_ (**A**), PM_2.5_ (**B**), NO_2_ (**C**), and SO_2_ (**D**). RR, rate, ratio.

**Table 1 ijerph-19-05988-t001:** Average daily concentrations of air pollutants, average daily values of meteorological variables, and total counts of asthma admission by city.

	Warsaw	Cracow	Tricity
PM_10_, μg/m^3^, median (IQR)	28.4 (20.7–40.3)	36.3 (24.4–60.0)	15.5 (11.2–22.6)
PM_2.5_, μg/m^3^, median (IQR)	20.6 (14.5–31.9)	26.0 (17.1–46.5)	11.5 (7.87–17.6)
NO_2_, μg/m^3^, median (IQR)	33.6 (26.1–41.4)	40.8 (33.6–49.0)	13.8 (9.69–19.1)
SO_2_, μg/m^3^, median (IQR)	4.56 (2.94–7.12)	5.63 (4.02)	2.61 (1.84–3.92)
Temperature, °C, median (IQR)	9.58 (2.90–16.9)	9.90 (2.95–16.6)	9.29 (3.71–5.9)
Relative humidity, %, median (IQR)	77.5 (66.1–87.1)	78.2 (68.6–86.2)	74.9 (67.6–80.7)
Atmospheric pressure, hPa, median (IQR)	1003 (998–1008)	988 (983–992)	1007 (1001–1012)
J45 and J46 admissions, *n*	17,015	8062	6842
J45 and J46 admissions per million, *n*	9452	10,470	9122

IQR, interquartile range.

## Data Availability

Data are available from the corresponding author.

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
