# Peer review of "Ambient Air Pollution and Risk of Admission Due to Asthma in the Three Largest Urban Agglomerations in Poland: A Time-Stratified, Case-Crossover Study"

_ijerph, 2022, doi:10.3390/ijerph19105988_

Round 1

Reviewer 1 Report

This article studies the association between concentrations of air pollutants and hospital admission in three urban agglomerations in Poland. In general, this is an interesting work because no similar studies have been carried out in Poland. However, the relationship between outdoor air pollutant concentrations and hospital admission may not offer an accurate understanding of the influence of air pollution on asthma, given that the proportion of asthma patients with wild and moderate exacerbations is high. Authors should at least include this shortcoming in the discussion.

Other comments:

L31: “In 2016, an estimated…”. Please double-check and rewrite the sentence.

L34: The meaning of “an interaction of genetic and environmental factors” is unclear. Please rewrite it.

L43: “nitric dioxide” -> nitrogen dioxide.

L107: “cross-bases” -> cross-basis.

L126: “Fig 1D” -> Figure 1E.

Author Response

Dear Reviewer,

On behalf of the Co-Authors and myself I would like to thank you for reviewing our manuscript. All of the responses to your valuable comments you could find in the attachment. I would like to kindly inform you, that we have revised the manuscript according to the remarks given by all of the Reviewers, thus the indicated (tracked) were made in response to all review reports.

Yours faithfully,

Artur Badyda.

Reviewer 2 Report

Comments to the authors:

Abstract

Page 1, line 13, “but data for Poland are lacking”- I thin these words are not necessary.

Page 1, line 15, “of various air pollutants”. Please mention the pollutants name here inside a parenthesis.

Page 1, line 16, is it “distributed lag non-linear model (DLNM) ?

Page 1, line 18, rate ratio, 95% confidence interval (CI)

Introduction

Starting of the introduction should be improved. The authors' research question was on the general population. So they should start with the asthma disease in the general population. Then they may mention other types of population groups after that. In line 32, if the authors mention the disease burden, they should present any DALYS or global burden of disease data on asthma worldwide, but especially in Poland. In line 48, “worst in Europe”- presents some specific data here.

Materials and Methods

2.1 Setting: It would be better if the authors may present the location with a map.

2.2 Statistical analysis: In line 97, please check the term- “distributed lag linear-nonlinear models (DLNM) ”

Results

3.1 Air pollution and asthma hospitalizations

The authors may present the table as:

Median (IQR)

Warsaw

Cracow

Tricity

IQR, Interquartile ranges; n, counts

Additional comments:

>> The authors could perform second-stage modeling which means combining the city-specific results (more specifically meta-analysis).

>> The authors could present sensitivity and sub-group analysis in this manuscript.

Author Response

Dear Reviewer,

On behalf of the Co-Authors and myself I would like to thank you for reviewing our manuscript. All of the responses to your valuable comments you could find in the attachment. I would like to kindly inform you, that we have revised the manuscript according to the remarks given by all of the Reviewers, thus the indicated (tracked) changes were made in response to all review reports.

Yours faithfully,

Artur Badyda.
